# The apple detection method based on multimodal features

**Xiaoyang Liu** [1]*, **Chongyang Hu**[1], **Xupeng Huang**[1], **Chenxin Sun**[1], **Rongjin Zhu**[1], **Cheng Wang**[1], **Yuxiang Zhang**[1], **Qian Shen**[1], **Hongbiao Zhou**[1], **Chengzhi Ruan**[2,3]

1 Faculty of Automation, Huaiyin Institute of Technology, Huaian, China, 2 The Key Laboratory for Agricultural Machinery Intelligent Control and Manufacturing of Fujian Education Institutions, Wuyishan, China, 3 School of Mechanical and Electrical Engineering, Wuyi University, Wuyishan, China

☯ These authors contributed equally to this work.

* leoliuxy@foxmail.com

**Data availability statement:** The relevant dataset is available at https://github.com/oxygenhcy/DATA.git.

## Abstract

Accurate detection of apples and other fruits in complex environments remains a formidable challenge due to the intricate interplay of varying lighting conditions, occlusions, and background clutter. Traditional detection methods, which primarily rely on RGB images or incremental improvement of deep learning models, often fail to achieve satisfactory detection accuracy. To address this, an innovative method of apple detection is proposed to improve the detection performance through multimodal feature fusion rather than radical architectural modifications. The proposed method integrates four complementary modalities: RGB image, color and edge feature maps, depth feature map, and point clouds. Chromatic properties of fruits and geometric boundaries of fruit-tree structures are captured by color and edge feature maps extracted from RGB inputs, which are weighted and fused into a composite feature channel. The depth map and point clouds acquired via binocular active infrared stereo cameras provide additional spatial information. The depth feature image is used as a standalone feature channel. Given the significant modal discrepancies between point clouds and RGB data, a preprocessing pipeline is implemented: voxel sampling and local anomaly detection are first applied to denoise and fill holes in point clouds, followed by recalibrated mapping to ensure spatial alignment with RGB image. The XYZ coordinates of processed point clouds are then used as three distinct feature channels. Finally, the YOLOv5 input layer is redesigned to accept multi-channel feature inputs.Multimodal fusion enriches the feature representation accessible to YOLOv5, enhancing model robustness against lighting variations and background noise. Experimental results demonstrate that the proposed method achieves 95.8% precision (P), 96.0% recall (R), and 95.9% F1-score in complex scenarios. Compared to baseline methods using RGB-only and RGB+depth inputs, precision improvements of 7.4% and 6.3% are observed respectively.

**Funding:** This work was supported by the following funding agencies: Xiaoyang Liu (XL): ● National Nature Science Foundation of China (Grant Nos. 32301711 and 61903288) [https://www.nsfc.gov.cn/] ● Natural Science Research Foundation of Higher Education Institutions in Jiangsu Province (Grant No. 21KJB210018) [https://jyt.jiangsu.gov.cn/] Chengzhi Ruan (CR): ● Open Project Program of the Key Laboratory for Agricultural Machinery Intelligent Control and Manufacturing of Fujian Education Institutions (Grant No. AMICM202201) [https://www.wuyiu.edu.cn/jdxy/njznkzyzzjsfjsgxzdsys/list.htm/] ● Natural Science Foundation of Fujian Province (Grant No. 2021J011132) [http://kjt.fujian.gov.cn/] For the funding associated with Xiaoyang Liu (XL): The funders had a role in the study design, data collection and analysis, decision to publish, or preparation of the manuscript. For the funding associated with Chengzhi Ruan (CR): The funders had no role in study design, data collection and analysis, decision to publish, or preparation of the manuscript. There was no additional external funding received for this study.

**Competing interests:** The authors have declared that no competing interests exist.

## 1 Introduction

With the rapid development of precision agriculture, the fruit detection technology in large-scale orchards plays a crucial role in fruit harvesting and quality assessment. According to data from the Chinese National Bureau of Statistics, China's apple production in 2024 was 51.2851 million tons that ranked first in the world[1]. Therefore, the demand for automated harvesting and quality assessment is increasing. However, the adaptability of existing image recognition technology in complex environments remains significantly lacking. This limitation restricts its practical effectiveness in agricultural scenarios.

Traditional fruit recognition methods use artificial features like texture, color, and shape. For example, researchers combined these features with models to identify apple parts [2,3]. However, these methods rely on manually designed features and struggle in complex environments. Changes in lighting can distort colors and textures, leading to uncertain segmentation results, such as misclassifying shadows as fruit regions or producing incomplete outlines.

With the development of deep learning technology, scholars have applied deep learning to detect fruits widely [4–7]. The YOLO (You Only Look Once) series is one of the popular directions in deep learning. In the researches of fruit detection, a variety of improvement methods for the YOLO model have been proposed to enhance model performance or adapt to research tasks, which mainly involve model simplification, addition and replacement of modules et al. For example, Zhao et al. [8] used the object detection framework of YOLOv3(You Only Look Once version 3) to recognize apples under different light environments and improved detection speed by simplifying the backbone network. Wang et al. [9] improved the YOLOv5(You Only Look Once version 5) algorithm by replacing the standard convolutional module with the inverted residual convolutional module in MobileNetv2(MobileNet version 2) and introduced the least squares method to the misjudgments of correct model to fast recognize and track apple fruits. This method optimizes the detection of associative objects, reduces model size and improves detection speed. In addition, the adjustment or introduction of improved loss functions based on existing or self-developed models are also common optimization methods of deep learning. Kang and Chen[10] developed an apple detection framework called "LedNet". To achieve fast and robust real-time detection of apples in orchards, this framework integrates an automatic label generation module, employs feature pyramid networks and atrous spatial pyramid pooling to enhance detection performance, and designs a lightweight backbone network to improve computational efficiency. Chen et al.[11] constructed a dataset for apple tree organ segmentation (branches, buds, leaves, and their connecting parts) during the bud stage and proposed an improved YOLOv8(You Only Look Once version 8) - based solution. To enhance the accuracy and precision of apple tree organ segmentation in complex natural environments effectively, the model integrates the advanced convolutional network modules ConvNeXt V2(Convolutional Next Version 2), MSDA(Multi-Scale Dilated Attention), and DSConv(Depthwise Separable Convolution). The related researches above mainly improve recognition performance by the optimization and improvement of framework structure, which do not fully exert the strong learning power of deep learning and lack researches and analysis of the multimodal feature information[12,13].

With the rapid development of machine vision and sensor hardware in recent years, depth cameras like the Microsoft Kinect have been widely used in fruit and vegetable recognition and positioning research [14,15]. These technological advancements have enabled us to capture three-dimensional information of fruits and vegetables, which is a valuable data source for deep learning models. Deep learning has strong learning capabilities and can extract complex features from these data. However, it is difficult to fully exert the advantages of deep

learning because the information contained in color images is limited. To overcome this limitation, researchers have begun to explore methods to integrate other morphological features. Such integration has not only made progress in improving the accuracy of fruit recognition but also provided a new technical direction for agricultural automation[16,17]. For example, Liu et al.[18] proposed a multimodal hierarchical fusion method based on attention mechanism for instance segmentation of tomato stems in robotic vision system. This method distinguishes between leaves and fruits with similar colors by fusing RGB and near-infrared images effectively. Mao et al.[19] employed a cucumber recognition method that integrates multiple features, to enhance the recognition accuracy of cucumbers against complex backgrounds significantly. This method employs the I-RELIEF(Iterative RELIEF) algorithm to select key features and fuses features extracted by deep learning.

Although multimodal fusion strategies perform well in complex environments such as orchards, how to select, obtain, and fuse related features to achieve efficient recognition of fruits is still a technical challenge. In addition, in order to improve the level of agricultural automation and enhance the operational accuracy of robot picking systems, providing accurate spatial information and fruit location remains a significant challenge.

In response to the above challenges, a new multimodal fusion method based on the YOLOv5 object detection framework for efficient recognition is proposed. This method significantly enhances model performance by integrating RGB images, key fruit feature map, depth feature, and point clouds at the input end. The key fruit feature is obtained by the weighted fusion of the red-green difference and edge feature maps. This strategy of multimodal fusion enhances the model's robustness to different lighting conditions and background noise and recognizes apples more efficiently.

## 1.1 Our contribution

The main contributions of this study include:

1. Proposing an innovative multimodal fusion method that integrates RGB images, key fruit feature map, depth feature, and point clouds to enhance the performance of the object detection model.
2. Developing a weighted fusion technology of key fruit features based on red-green difference and edge feature maps that reduces the redundancy and interference of features;
3. The study not only has application value in the agricultural field, but its multimodal fusion technology also provides new ideas for object detection in other fields.

## 2 Materials and methods

In this research, the apple images were captured by the Intel® RealSense™ Depth Camera D455 that is a member of the Intel RealSense D400 series and renowned for its versatility in both indoor and outdoor settings. The D455 depth camera can capture high-resolution RGB and depth images. The resolution of depth images reaches up to 1280×720 and the resolution of color images reaches up to 1280×800. In addition, the D455 depth camera can also capture point cloud images and three-dimensional point cloud data. Given the excellent environmental adaptability and functionality of this camera, it was selected for the relevant data collection work in this study.

### 2.1 Data collection

In this research, the apple images were collected from the outdoor apple orchard named Matou in Huai'an City, Jiangsu Province. To enhance the variety of apple images, some apple

images were taken using apple models (these models closely resemble real apples in appearance and texture). The D455 depth camera was employed to capture all images from diverse perspectives and distances. Fig 1a is the color image from the camera's color sensor with a resolution of 848 × 480 pixels. The images are stored in RGB8 color format (RGB8 is an 8-bit color format for each red, green, and blue channel). Fig 1b is the pseudo-colored depth image of Fig 1a with the same resolution. The depth images are formatted in Z16 (Z16 is a 16-bit format for storing depth information). Additionally, the depth image is converted into a pseudo-colored image and visualized using the color mapping function in the RealSense SDK. The depth information ranges from 0 to 4 meters.

As illustrated in Fig 2, the point cloud image corresponding to the color image in Fig 1a is generated via the integration of 30 image frames, which were captured by the camera under program control. This program enables the camera to acquire the 30 frames near-simultaneously, ensuring all frames correspond to the identical 3D spatial scene. Conventionally, point cloud images are derived from single-frame data; however, this approach often results in the omission of key components within the point cloud due to insufficient data sampling. To address this limitation, the present study adopted the aforementioned multi-frame accumulation strategy: given the near-simultaneous acquisition of the 30 frames and their consistent mapping to the same 3D spatial domain, all 3D points extracted from the 30 frames are directly aggregated into a unified point cloud dataset. For points that overlap in 3D spatial coordinates across frames, they are retained as a single point without introducing interference; meanwhile, non-overlapping points supplement the spatial information coverage. This method effectively enhances the density and spatial coverage of the point cloud, thereby complementing the key components of the point cloud image and improving its overall completeness.

In an effort to improve the accuracy of apple recognition under natural light conditions, a dataset has been enriched with images of apples captured across a range of light intensities (as illustrated in Fig 3 and organized alphabetically by light intensity). Furthermore, the dataset includes images of apples taken from various angles and distances. This diversification of the dataset characteristics serves to mitigate the risk of model overfitting and enhances the model's generalization capabilities.

## 2.2 Data processing

Although multi-frame accumulation can improve the density of the point cloud, it also introduces more noise. Therefore, this research conducted noise reduction and filling treatment

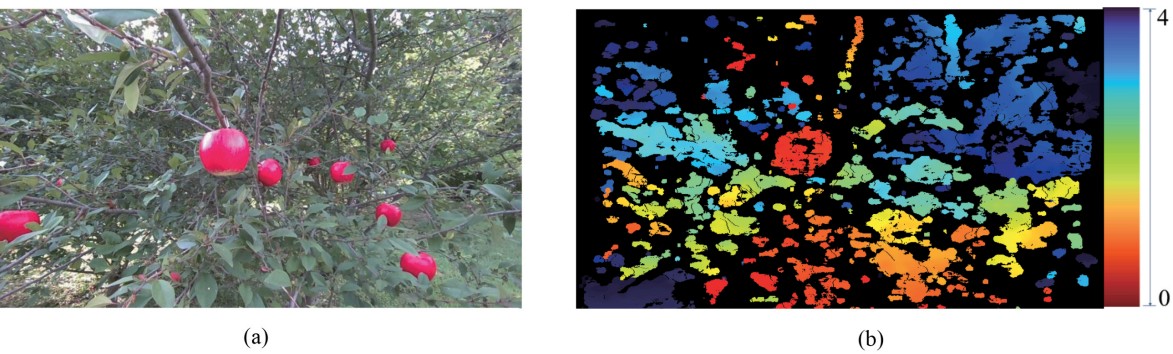

(a)  (b)

**Fig 1. Apple color image and pseudo-color depth view set.** (a) Color image of apple, and (b) Apple pseudo-color depth image.

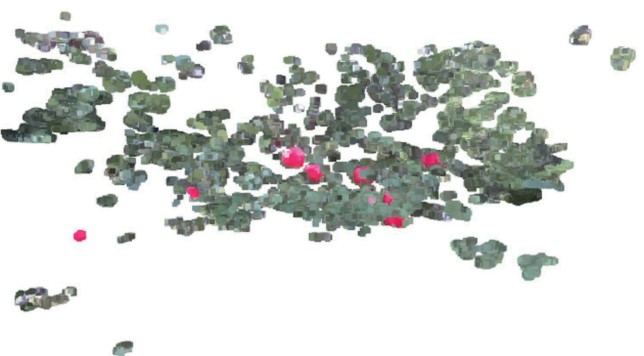

**Fig 2. Apple point cloud generated by python script.**

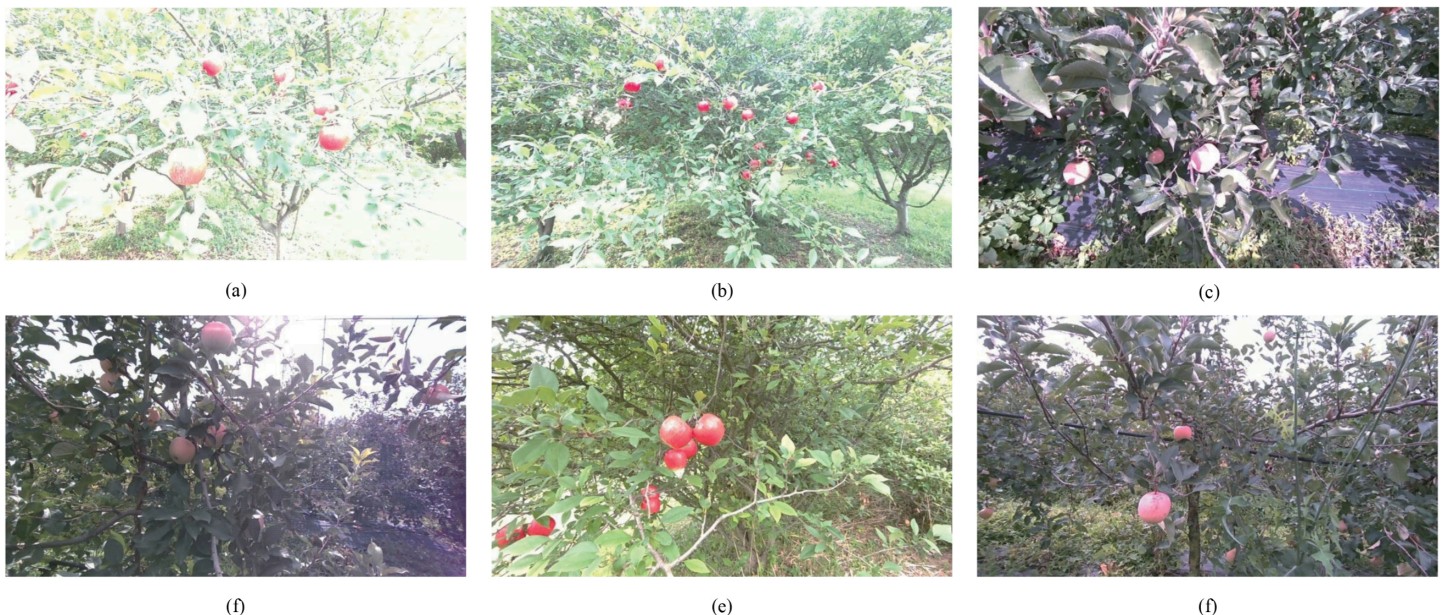

| (a) | (b) | (c) |
| (f) | (e) | (f) |

**Fig 3. Apple images captured under different light intensity conditions.** (a) Apple image under light intensity a, (b) Apple image under light intensity b, (c) Apple image under light intensity c, (d) Apple image under light intensity d, (e) Apple image under light intensity e, (f) Apple image under light intensity f.

on the point cloud. This research employs Voxel Down Sampling technology to reduce point cloud noise. This technique discretizes the point cloud into a voxel grid (each voxel representing a small spatial cube). By retaining the first point in each voxel, the number of points is reduced while preserving the overall shape of the point cloud. To further reduce noise, this research also utilize a local outlier detection method to identify and remove outliers. Fig 4 illustrates the flowchart of the local outlier detection method that focuses on analyzing the distribution of distances of points within the local neighborhood of each point and identifies outliers based on the standard deviation. The specific implementation process is as follows: First, select 30 nearest neighbor points within the neighborhood of each point in the point cloud. Then, calculate the distance from each point to these neighborhood points and further calculate the standard deviation of these distances. When setting the threshold, the average distance plus 2 times the standard deviation is used as the criterion for identifying

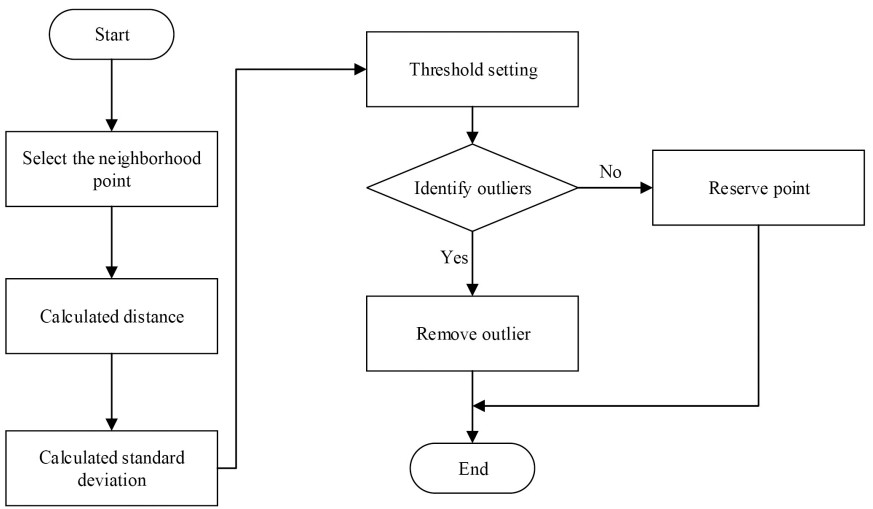

**Fig 4. Flowchart of the local outlier detection method.**

outliers (this threshold is based on the $3\sigma$ principle of normal distribution, which means that approximately 99.7% of data points are within 3 standard deviations of the mean). If the average distance of a point to its neighborhood points exceeds this threshold, the point is marked as an outlier. Finally, remove these identified outliers to address the excessive noise caused by multi-frame accumulation.

After noise removal, zeros are applied to fill empty regions in the point cloud to ensure data integrity. Finally, due to the inconsistency between the coordinate system of the D455 depth camera and the real-world coordinate system, a coordinate transformation is applied to the point clouds before saving the point cloud image to ensure it conformed to the scale of the real world. Through the aforementioned operations, a high-quality point cloud image can be obtained. In this research, high-quality point cloud images contribute to obtaining more accurate spatial feature information, which is crucial for multimodal feature extraction. Specifically, such accurate spatial features ensure precise alignment between 3D and 2D visual data in multimodal fusion, preventing target missed or false detection and supporting better target detection accuracy.

After the completion of image collection and a rigorous screening process, a total of 572 images are meticulously selected for the experiment. These images accompanied by their respective depth and coordinate feature data are consolidated into a dataset designated as DA1. The depth and coordinate features are extracted from the gathered depth image data and point cloud image data, respectively. To refine the model's training and evaluation procedures, the DA1 dataset is meticulously partitioned into training, validation, and testing subsets with the proportions of 70%, 10%, and 20%, respectively. Additionally, data augmentation techniques are implemented on the training subset, which serves to bolster the model's generalization and reduce its susceptibility to overfitting. This involve the application of diverse transformations, such as rotation, scaling, cropping, color adjustment and the incorporation of noise, which enhanced the diversity of the dataset. These methodologies provide a more comprehensive set of learning materials for the deep learning model, which enhanced its performance and stability in real-world scenarios.

YOLOv5 is a supervised deep learning approach, and its training and validation dataset require the addition of precise labels. For this research, the open-source image annotation

tool, LabelImg, is utilized for manual annotation in both the training and validation datasets. During the annotation process, fruits with heavy occlusion are omitted, specifically when the obscured area of a fruit exceeds 80% of its total area. This measure is designed to maintain the integrity of the fruits, which in turn enhances the model's training efficacy and predictive accuracy.

In this research, dataset DA1 encompasses color image data, depth image data, and point cloud data. To ensure the alignment of corresponding color and depth frames, we acquire color and depth image data using a consistent script, identical resolution and fixed position. Nevertheless, the point cloud data is synthesized from an accumulation of frames. Despite a fixed camera position during the collection process, minor perturbations may arise from external factors. Therefore, it is important to validate the alignment between color image data and the corresponding point cloud data. This verification process involving a paired comparison of color image data and point cloud data is conducted on the PyCharm Community Edition platform and encompasses the subsequent steps: Data Preparation Phase; Iterative Projection Mapping; Obtaining Pixel Coordinates; Visualization and Result Verification.

**2.2.1 Data preparation phase.** A set of color images and point cloud images was randomly selected from the DA1 dataset, as illustrated in Fig 1a and Fig 2, respectively. The procedure involves initializing the depth camera and retrieving the intrinsic parameters, which include *fx*, *fy*, *cx*, and *cy*. Here, *fx* and *fy* represent the focal lengths along the x-axis and y-axis of the camera respectively, which are the distances from the imaging plane to the projection center. Meanwhile, *cx* and *cy* denote the coordinates of the principal point on the x-axis and y-axis in the image coordinate system respectively. The principal point is the location of the optical center in the image. The selected color images and point cloud images from the same group are loaded into the system. This step involves reading the image files to prepare them for further processing and analysis.

**2.2.2 Iterative projection mapping.** To achieve the projection from 3D to 2D, the initial step is to transform the point cloud coordinates into camera coordinates:

$$P_{cam} = E \cdot \begin{bmatrix} x \\ y \\ z \\ 1 \end{bmatrix} \tag{1}$$

$$E = \begin{bmatrix} 1 & 0 & 0 & 0 \\ 0 & 1 & 0 & 0 \\ 0 & 0 & 1 & 0 \\ 0 & 0 & 0 & 1 \end{bmatrix} \tag{2}$$

In this research, the matrix $E$ represents the extrinsic parameters and is initialized as the identity matrix; $P_{cam}$ denotes the 3D points in the camera coordinate system, and $[x\ y\ z\ 1]$ represents the homogeneous coordinates of the 3D points in the point cloud.

Subsequently, the process involves projecting the points from the camera coordinate system onto the image plane. In this context, it is necessary to eliminate the last component of the homogeneous coordinates, utilizing solely the first three components of $P_{cam}$ for the projection, which are denoted as $P_{cam,3}$:

$$P_{cam,3} = \begin{bmatrix} x_{cam} \\ y_{cam} \\ z_{cam} \end{bmatrix} \tag{3}$$

In this context, $x_{cam}$, $y_{cam}$, and $z_{cam}$ represent the coordinates of a point in the camera coordinate system.

**2.2.3 Iterative projection mapping.** Subsequently, the processed camera coordinates are transformed into pixel coordinates using the camera's intrinsic parameter matrix:

$$P_{pixel} = K \cdot P_{cam,3} \tag{4}$$

$$K = \begin{bmatrix} f_x & 0 & c_x \\ 0 & f_y & c_y \\ 0 & 0 & 1 \end{bmatrix} \tag{5}$$

In this context, $K$ denotes the intrinsic parameter matrix of the camera, where $f_x$ and $f_y$ are the focal lengths along the x-axis and y-axis, respectively; $c_x$ and $c_y$ are the coordinates of the camera's optical center; $P_{cam,3}$ represents the first three components of a 3D point in the camera coordinate system; and $P_{pixel}$ denotes the pixel coordinates of the point on the image plane. Subsequently, $P_{pixel}$ is normalized to obtain the actual pixel coordinates:

$$\begin{bmatrix} u \\ v \\ 1 \end{bmatrix} = \frac{1}{z_{cam}} \begin{bmatrix} x_{cam} \cdot f_x + c_x \cdot z_{cam} \\ y_{cam} \cdot f_y + c_y \cdot z_{cam} \\ z_{cam} \end{bmatrix} \tag{6}$$

In this process, $u$ and $v$ are the pixel coordinates on the image plane after transformation by the intrinsic matrix and normalization, corresponding to the horizontal and vertical coordinates of the image respectively.

Due to the nature of the projection mapping being a mirror mapping, the image is subjected to a mirror-flipping operation:

$$u' = m_w - 1 - u \tag{7}$$

In this context, $u'$ represents the horizontal coordinate after the mirror-flipping operation, and $m_w$ denotes the width of the image in pixels. Finally, the projection offset is corrected:

$$\begin{cases} u_o = u' + x_{offset} \\ v_o = v + y_{offset} \end{cases} \tag{8}$$

In this context, $x_{offset}$ and $y_{offset}$ are the correction parameters used to adjust for the projection offset, representing the displacements in the x-axis and y-axis directions respectively; $u_o$ and $v_o$ are the pixel coordinates on the image plan e after correction, corresponding to the accurate horizontal and vertical coordinates of the image respectively.

**2.2.4 Visualization and result verification.** This research employs visual verification to provide an intuitive experimental outcomes. In Fig 5a, the green boxes represent the fruit regions that have been manually annotated. Then, the experiment utilizes projection mapping to annotate the selected regions on the point cloud image in black. This process serves to validate the precision of the spatial coordinate position data associated with the key regions of the color image. Fig 5b, Fig 5c, and Fig 5d respectively present the front, right, and top views of the experimental validation results. It is evident that the key regions identified in the RGB color image are marked in black on the point cloud image. This also fully verifies the accuracy of the spatial coordinate position data corresponding to the color images (Note that the presence of red points in the point cloud does not indicate inaccuracy. These points are due to the

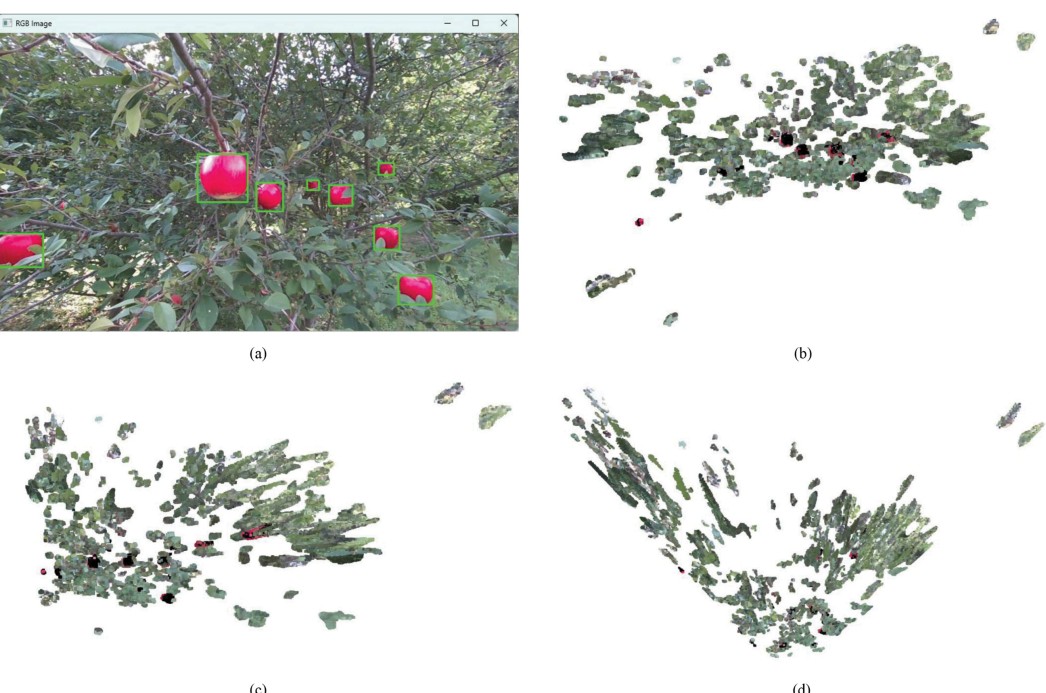

**Fig 5. Manually annotated and 2D-to-3D mapped views of apples.** (a) Manually Annotated Apple Image, (b)2D-to-3D Mapped Apple Point Cloud Front View, (c) 2D-to-3D Mapped Apple Point Cloud Right View, (d) 2D-to-3D Mapped Apple Point Cloud Top View.

use of a single depth camera, which results in a scattered point cloud visualization, and the adoption of a multi-frame capture method. Some red points are located in sparse areas that are more susceptible to noise. The verification of data accuracy is based on whether the black points adequately cover the central part of the key regions and their surrounding diffusion areas).

## 2.3 Fruit feature extraction

Traditionally validated and effective fruit features may not be readily learned by deep learning models. Therefore, integrating these features directly into the input layer not only enriches the set of input features but also reduces the uncertainty in training, ensuring that these proven features are effectively captured and learned by the model. The initial composition of dataset DA1 includes color image data, depth image data, and point cloud image data. To incorporate traditional fruit feature information into the input layer of the object detection model, DA1 needs to be expanded further. Beyond the original image's R, G, B color channels, additional feature dimensions are imperative to construct a multi-channel image. This measure is designed to enrich the input data for the object detection algorithm. The selection of fruit features is guided by two fundamental criteria: firstly, the features should be able to significantly differentiate the fruits from the background; secondly, the features should be graphically representable to ensure seamless integration with the original RGB color channels. This multi-dimensional feature fusion strategy enhances the recognition accuracy and robustness of the fruit features within the object detection model.

In the field of fruit feature analysis, color features play an important role particularly in detecting red, orange, and similarly-hued fruits. The R-G color difference operator, which

is a classic tool for detecting red fruits such as apples, significantly improves the color contrast between the fruit and its background. The feature map of color difference is an intuitive graphical representation of color features. By highlighting luminance disparities among regions in an image, it effectively separates fruits from the background. As depicted in Fig 6d, Fig 6e, Fig 6f, Fig 6p, Fig 6q, and Fig 6r, the background elements, such as the sky, branches and leaves, are reduced effectively in the R-G color difference map of the apple image while the red regions of the fruit are enhanced, which distinguishes fruits significantly from the surroundings. However, some parts of fruits may be missed in complex lighting.

In this research, the textural feature is not adopted because the images are captured in orchards with wide view and the textures are not clear. Concurrently, despite the apple's shape being approximately circular, this feature is challenging to represent intuitively in the image. Consequently, this research employs edge feature map to articulate the texture and shape features of the apple. After evaluating multiple edge detection operators, the Laplacian operator demonstrates superior accuracy and coherence in generating edge images. Notably, in high-contrast settings, it more precisely captures the contour features of apples.

Therefore, the original color apple images are converted to grayscale, and then the Laplacian operator is used to identify the edges in the images. The operator accurately identifies edge locations based on the second-order derivative changes in image brightness, and these locations typically coincide with the contours of objects within the image. As depicted in Fig 6g, Fig 6h, Fig 6i, Fig 6s, Fig 6t, and Fig 6u, the edge feature images display a significant contrast between the apple and its background. The edges of the apple are relatively sparse and indicates the smooth surface characteristics of apples. In contrast, the branches and leaves in the background exhibit a complex texture attributed to the dense arrangement of edge lines. The distribution of these edges lines not only reveal the diversity of the image textures but also imply the morphological features of the fruit.

After acquiring diverse fruit feature data, the collected information is integrated with the dual goals of conserving computational resources and preserving key fruit feature details. As shown in Fig 6j, Fig 6k, Fig 6l, Fig 6v, Fig 6w, and Fig 6x, the color difference map and edge map are subjected to initial normalization before being merged in a weighted ratio of 6:4. This fusion process aims to extract the key fruit feature information map. The key fruit feature data is integrated into the object detection model, which establishes a robust data foundation for subsequent image recognition and object localization tasks.

## 2.4 Multimodal image fusion

Building on the original DA1 dataset, the key fruit feature are integrated to form a new dataset, designated as DA2. The process of merging the feature image data with the original image data is detailed in Fig 7 and divided into two main stages:

Step 1: Without affecting the existing annotation information, the color image is initially decomposed into its R, G, and B color components. Subsequently, the R-G operator is employed to compute the disparity between the red and green channels, resulting in a color difference map. Concurrently, the color image is converted to grayscale, and edge detection is performed using the Laplacian operator to obtain an edge map. These color difference map and edge map are merged based on a predefined weight ratio to generate a key fruit feature map.

Step 2: The point cloud image is aligned with the RGB image to extract point cloud coordinate data (XYZ) that corresponds to the pixel coordinates of the RGB image. Finally, the R, G, B channels, key fruit feature map, depth information and point cloud coordinate data (XYZ) are combined to create an image dataset with 8 channels.

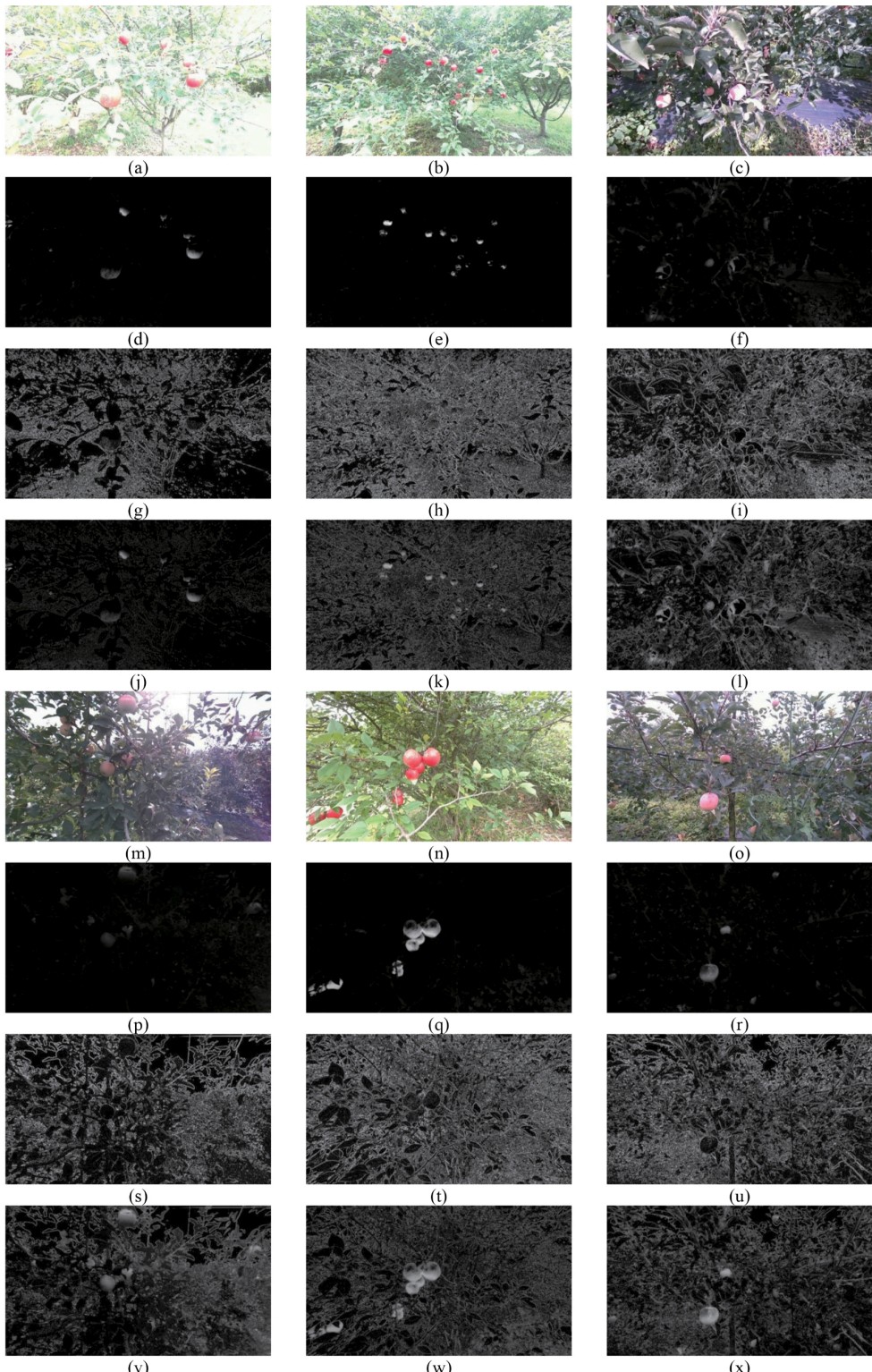

**Fig 6. Apples under varying illumination intensities: Chromaticity, edge, and key fruit feature maps.** (a) Original apple image under lighting condition a, (b) Original apple image under lighting condition b, (c) Original apple image under lighting condition c, (d) Chroma image of apple under lighting condition a, (e) Chroma image of apple under lighting condition b, (f) Chroma image of apple under lighting condition c, (g) Edge image of apple under lighting

condition a, (h) Edge image of apple under lighting condition b, (i) Edge image of apple under lighting condition c, (j) key fruit characteristic mapping of apples under light a, (k) key fruit characteristic mapping of apples under light b, (l) key fruit characteristic mapping of apples under light c, (m) Original apple image under lighting condition d, (n) Original apple image under lighting condition e, (o) Original apple image under lighting condition f, (p) Chroma image of apple under lighting condition d, (q) Chroma image of apple under lighting condition e, (r) Chroma image of apple under lighting condition f, (s) Edge image of apple under lighting condition d, (t) Edge image of apple under lighting condition e, (u) Edge image of apple under lighting condition f, (v) key fruit characteristic mapping of apples under light d, (w) key fruit characteristic mapping of apples under light e, (x) key fruit characteristic mapping of apples under light f.

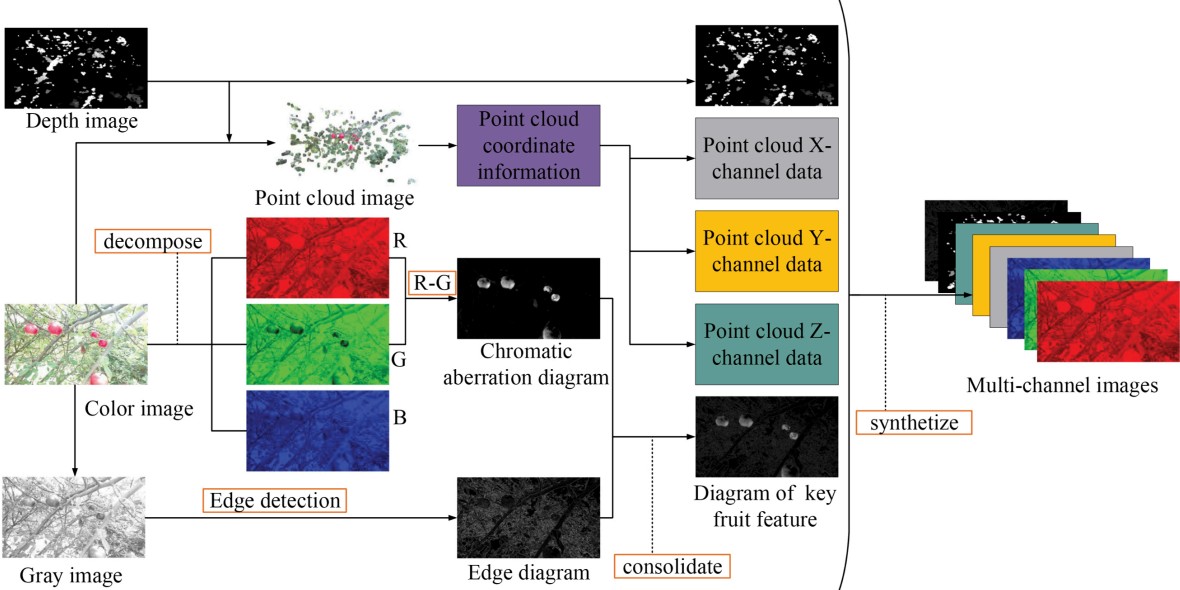

**Fig 7. The flow chart of synthesizing the multi-channel images.**

## 2.5 Deep learning model

YOLOv5, which is a traditional single-stage object detection framework, facilitates rapid and accurate object detection by directly extracting features through convolutional neural networks and subsequently identifying and localizing objects. Compared to traditional two-stage detection frameworks, YOLOv5 simplifies the detection process, reduces the consumption of computational resources, and maintains high detection accuracy. In the context of this research, a refined version of YOLOv5 is employed for the recognition and localization of apples in natural environments, aiming to achieve real-time and efficient object detection.

To further integrate more features and enhance the network's adaptability to multi-modal fruit data, targeted architectural modifications are made to the original YOLOv5 framework, with the core improvement focused on optimizing the input layer structure as depicted in Fig 8. The original YOLOv5 input layer only supports 3-channel RGB images, so the modified input layer is first adjusted to an 8-channel configuration to accommodate multi-source feature data. These 8 channels include conventional RGB images, key fruit features(F), depth feature information (D), and point cloud coordinates information (XYZ), matching the 8-channel image configuration of dataset DA2 used in this research.

Meanwhile, adjustments are made to the convolutional kernel of the input layer, whose dimensions follow the form W×H×m×n. Here, W×H denotes the kernel's length and width, m represents the number of kernel layers (matching the input image channels), and n is the

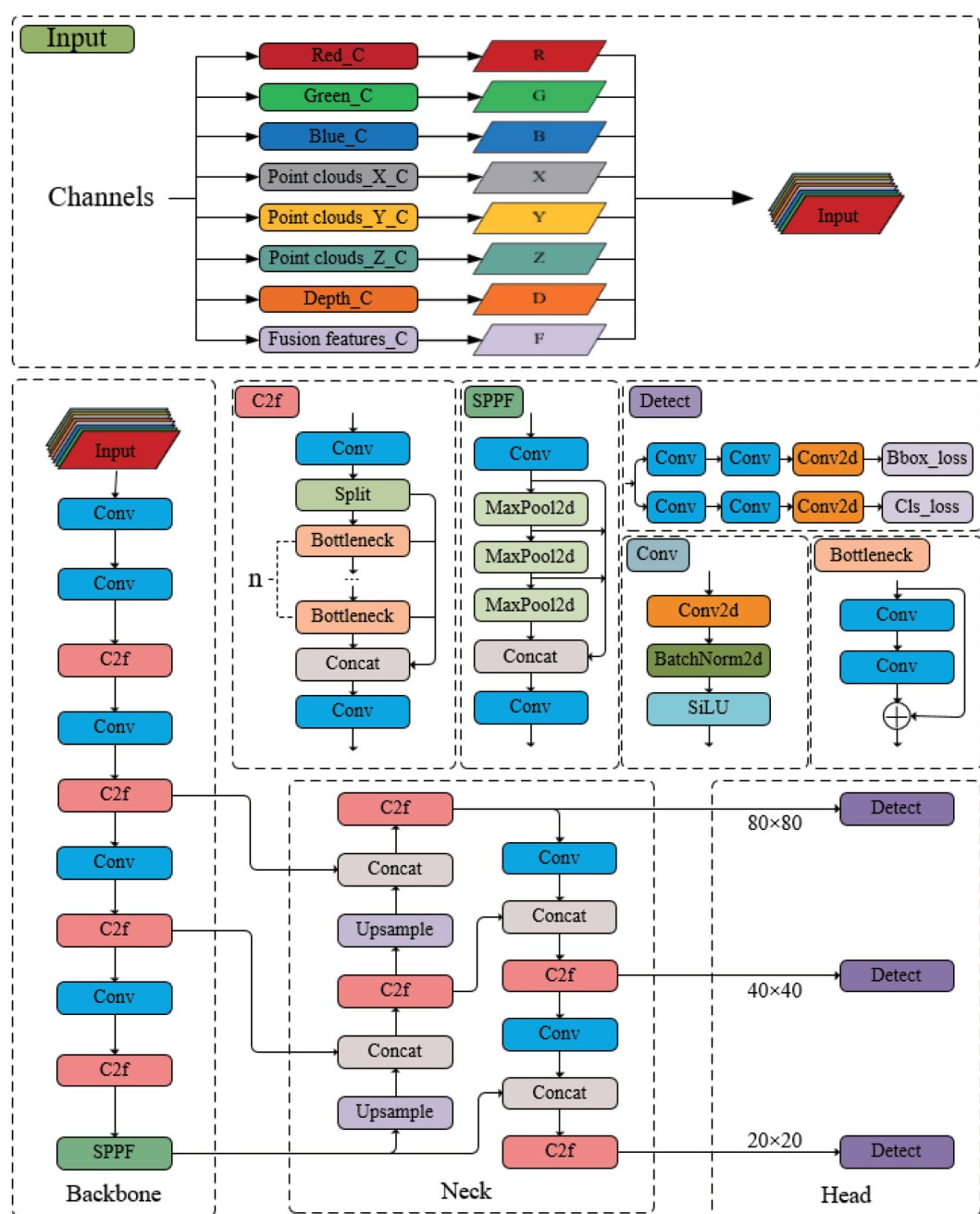

**Fig 8. Architecture of the YOLOv5 object detection model with optimized input layer.**

number of distinct convolutional kernels for feature extraction. In the original YOLOv5 network, W×H is set to 6×6 and n is established at 32. To align with the 8-channel input, only the parameter m is recalibrated from 3 (for 3-channel RGB) to 8, while W×H and n remain unchanged to ensure the stability of basic feature extraction functions.

Additionally, considering the anchor box sizes, nine distinct dimensions are selected based on previous experience: 10×13, 16×30, 33×23, 30×61, 62×45, 59×119, 116×90, 156×198, and 373×326. But the efficacy of these anchor boxes has yet to be evaluated. The optimal anchor

points are determined by calculating the ratio of the width and height to the anchor points, followed by the computation of the proportion of anchor boxes surpassing the threshold (Anchors Above Threshold) and the best possible recall rate (Best Possible Recall) to assess the current anchor boxes. If the existing anchor boxes are deemed suitable for the dataset, no recalculation is necessary; otherwise, the k-means clustering method is employed to recalculate the anchor boxes. This method initially extracts the width and height of all annotated boxes from the training dataset and then employs the k-means algorithm to cluster these dimensions, thus determining the optimal anchor box sizes. To enhance the precision of the clustering, data normalization is conducted, and the iteration count is set to 30. Following the initialization of the anchor boxes, a genetic algorithm is subsequently used to optimize these anchor boxes, enhancing the model's detection capability for small-sized objects. The genetic algorithm iteratively enhances the anchor box sizes through random mutation and fitness evaluation, with the fitness function predicated on the ratio of the anchor box's width and height to that of the true box. By establishing a threshold, this method effectively sieves out overly diminutive anchor boxes, ensuring the quality of the anchor boxes and bolstering the model's performance. Ultimately, this methodology enables the automatic adjustment of anchor box sizes to better align with the dataset's characteristics, thereby enhancing the precision and efficiency of object detection. The comprehensive process of the automatic optimization of YOLOv5 model anchor points is delineated in Algorithm 1 (Fig 9).

---

**Algorithm1:** AutoAnchor for YOLOv5 Model

**Require:** Dataset path, number of anchors (n), image size (img_size), threshold (thr), generations for genetic algorithm (gen), verbosity (verbose)
**Ensure:** Optimized anchor sizes

1:  **Begin**
2:  Load dataset from dataset path or create a loaded dataset object.
3:  Set the number of anchors (n), image size (img_size), threshold (thr), generations (gen), and verbosity (verbose).
4:  Initialize the anchor list (anchors) with predefined anchor sizes if not provided.
5:  Calculate the image shapes by scaling the image size to the dataset shapes.
6:  Generate random scales for data augmentation.
7:  Extract width and height (wh) from the dataset labels and apply the random scales.
8:  Define a metric function to compute the ratio of wh to anchors and find the best match.
9:  **If** the current anchors fit the dataset well (based on the Best Possible Recall threshold):
10:    Print the current anchors and their metrics.
11: **Else:**
12:    Use k-means clustering to find new anchor sizes from the dataset.
13:    Initialize the anchor list with the new sizes.
14:    **If** genetic algorithm is enabled:
15:       Perform genetic algorithm to further optimize the anchor sizes.
16:       - Define a fitness function based on the ratio metric.
17:       - Mutate and evaluate the anchors over generations.
18:       - Select the best anchors based on the fitness function.
19:    **End If**
20:    Print the optimized anchors and their metrics.
21: **End If**
22: Return the optimized anchor sizes.
23: **End**

---

**Fig 9. Pseudocode for automatic anchor optimization in the YOLOv5 model.**

## 3 Results and discussion

To evaluate the performance improvements in fruit detection using the refined YOLOv5 model framework with different feature fusion strategies, a comprehensive series of experiments was conducted for comparative analysis. The key experimental settings are detailed in Table 1 below, ensuring clarity and accessibility for reference.

### 3.1 Model testing

To evaluate the performance of the trained models, each model to be tested undergoes evaluation. The models' performance metrics were quantified utilizing key parameters, including precision $P$ (as delineated in Equation 9), recall $R$ (as delineated in Equation 10), and the composite score $F_1$ (as delineated in Equation 11).

$$P = \frac{TP}{TP + FP} \tag{9}$$

$$R = \frac{TP}{TP + FN} \tag{10}$$

$$F_1 = \frac{2PR}{P + R} \tag{11}$$

In this context, TP denotes the number of correctly detected apple predictions or the number of apples correctly detected; FP represents the number of prediction boxes that did not correctly detect apples; TP+FP is the total number of prediction boxes; FN indicates the number of apples not detected, and TP+FN represents the total number of apples.

The YOLO series of models were selected as the benchmark due to their fast speed, ease of training, and suitability for real-time object detection tasks. To compare the recognition effects and performance of common object detection frameworks such as YOLOv3, YOLOv5, YOLOv6, and YOLOv8, the color image data from dataset DA2 were used to train and test all four models. The test results are presented in Table 2. It can be observed that among the commonly used object detection frameworks, YOLOv5 demonstrated the best performance in apple recognition when using the same version without special optimization. Apple recognition, being a real-time object detection task, was conducted using the 's' (small) version because it has high requirements for the number of parameters. As can be seen from the

**Table 1**. **Key experimental settings.**

| Category | Specific Configuration |
|---|---|
| Software Platform | PyCharm Community Edition |
| Hardware Setup | Intel(R) Core(TM) i5-11400H CPU; 16GB RAM; 1× RTX 3050 GPU (12GB dedicated memory) |
| Optimization Algorithm | Stochastic Gradient Descent (SGD) |
| Training Batch Size | 4 (constrained by hardware limitations) |
| Initial Learning Rate | 0.01 |
| Learning Rate Decay Factor | 0.01 |
| Momentum Coefficient | 0.937 (0.8 during warm-up phase) |
| L2 Regularization Factor | 0.0005 |
| Warm-up Phase | 3 epochs (bias learning rate fixed at 0.1 during this period) |
| Maximum Training Epochs | 200 |

**Table 2.** The test data of different object detection frameworks.

| Detection frames | P (%) | R (%) | $F_1$ (%) | Number of parameters(MB) |
|---|---|---|---|---|
| YOLOv3 | 79.7 | 99 | 88.31 | 18.9 |
| YOLOv5 | 88.4 | 98 | 92.95 | 13.6 |
| YOLOv6 | 84.9 | 97 | 90.55 | 31.4 |
| YOLOv8 | 86.7 | 97 | 91.56 | 14.6 |

table, under the same version, YOLOv5 also has the smallest number of parameters. Overall, YOLOv5 is very suitable as the benchmark model for this task.

## 3.2 Performance evaluation and analysis under multiple configurations

The performance evaluation of the YOLOv5 model with the integration of a single feature information in the input layer is shown in Table 3. It can be observed that the basic RGB configuration already demonstrates the model's strong performance in handling standard three-channel images. Subsequently, by introducing a single feature information to enrich the input features, the model performance is effectively enhanced. Among the configurations that integrate only one type of feature information, it is evident that the RGB+(R-G), RGB+D, RGB+E, and RGB+XYZ configurations all contribute to improving model performance. However, the model with the RGB+XYZ configuration exhibits superior performance in terms of accuracy, recall, and F1 score. This indicates that the introduction of coordinate feature information has a significant positive impact on enhancing model performance. The model with this configuration can better understand the position and spatial relationships of objects in the image, which is crucial for precise object localization.

Color features and edge features are both artificial features, and as shown in Table 3, they only provide a modest improvement in model performance. Therefore, considering the direct integration of these features into the input layer through the fusion of multiple features can lead to performance enhancement. As can be seen from Table 4, the model demonstrates better performance in terms of accuracy, recall, and F1 scores under the RGB+(R-G)+E configuration. However, considering the potential for information redundancy and interference that may arise from the direct fusion of traditional features, the fruit feature extraction section employs a weighted fusion of artificial features to obtain key fruit features. From Table 4, it is evident that the model under the RGB+F configuration shows improved performance in accuracy, recall, and F1 scores after processing. This approach retains the critical parts of traditional features while reducing information redundancy and mutual interference to a certain extent.

**Table 3.** Performance evaluation of YOLOv5 model with the addition of a feature in the input layer.

| ID | Input Layer | NC | P (%) | R (%) | $F_1$ (%) |
|---|---|---|---|---|---|
| 1 | RGB | 3 | 88.4 | 98 | 92.95 |
| 2 | RGB+(R-G) | 4 | 89.4 | 97 | 93.05 |
| 3 | RGB+D | 4 | 89.5 | 97 | 93.1 |
| 4 | RGB+E | 4 | 89.8 | 97 | 93.26 |
| 5 | RGB+XYZ | 6 | 91.2 | 97 | 94.01 |

NC, Number of Channels; RGB, Red, Green and Blue Feature Information; (R-G), Chromaticity Feature Information; D, Depth Feature Information; E, Edge Feature Information; XYZ, Coordinate Feature Information.

**Table 4. Performance evaluation of YOLOv5 model with direct and indirect integration of traditional features.**

| ID | Input Layer | NC | $P$ (%) | $R$ (%) | $F_1$ (%) |
|----|-------------|-----|---------|---------|-----------|
| 1 | RGB+(R-G)+E | 5 | 91.9 | 97 | 94.38 |
| 2 | RGB+F | 4 | 92.9 | 96 | 94.42 |

F, Key Fruit Feature Information.

In most cases, the model's performance has been significantly improved by incorporating various feature information into the input layer. However, the configuration RGB+(R-G)+E+D did not result in the anticipated performance enhancement; instead, it had a negative impact. This is because chromaticity features (R-G) and edge features (E) overlap with the inherent color and structural information of RGB images, while depth features (D) introduce spatial data that conflicts with these visual cues, creating redundant and interfering signals that confuse the model's feature learning. This outcome emphasizes the importance of feature selection and fusion strategies in model design. As can be seen from Table 5, the configuration RGB+XYZ+D+F demonstrated the best performance among the models with performance significantly higher than other configurations. This model does not overly rely on the high-dimensional features of the input data and is more adaptable to different data distributions and detection tasks, exhibiting better generalization capabilities. Considering performance, efficiency, and generalization ability, the configuration RGB+XYZ+D+F is recognized as the optimal configuration due to its outstanding performance on key performance indicators.

## 3.3 Performance evaluation of different object detection models

The performance comparison of different object detection models is presented in Table 6. All models are evaluated on the same dataset, with each utilizing an 8 channel input configuration of RGB+XYZ+D+F. It can be observed that the SSD + VGG16 model demonstrates a certain performance level, with precision, recall, and F1 scores reaching 86.6%, 78%, and 82.08% respectively. Subsequently, the Faster RCNN + VGG16 model shows improved performance compared to SSD + VGG16, with precision, recall, and F1 scores at 88.2%, 81%, and 84.45%. Moreover, the YOLOv5 (Improved) model exhibits superior performance among the three models. Its precision, recall, and F1 scores are 95.8%, 96%, and 95.9% respectively, which are significantly higher than those of the other two models. This indicates that the improved YOLOv5 model has remarkable advantages in object detection tasks under this multimodal input and dataset condition, being able to more accurately identify and localize objects, thus achieving better overall detection performance.

**Table 5. Performance evaluation of the YOLOv5 model with the incorporation of multiple features in the input layer.**

| ID | Input Layer | NC | $P$ (%) | $R$ (%) | $F_1$ (%) |
|----|-------------|-----|---------|---------|-----------|
| 1 | RGB+(R-G)+E | 5 | 91.9 | 97 | 94.38 |
| 2 | RGB+(R-G)+E+D | 6 | 88.6 | 97 | 92.61 |
| 3 | RGB+F+D | 5 | 93.3 | 97 | 95.11 |
| 4 | RGB+XYZ+F | 7 | 94.4 | 96 | 95.19 |
| 5 | RGB+XYZ+D | 7 | 94.6 | 97 | 95.78 |
| 6 | RGB+XYZ+D+F | 8 | 95.8 | 96 | 95.9 |

**Table 6**. **Performance comparison of different object detection models.**

| ID | Models | $P$ (%) | $R$ (%) | $F_1$ (%) |
|---|---|---|---|---|
| 1 | SSD+VGG16 | 86.6 | 78 | 82.08 |
| 2 | Faster RCNN+VGG16 | 88.2 | 81 | 84.45 |
| 3 | YOLOv5(Improved) | 95.8 | 96 | 95.9 |

Improved, The YOLOv5 version with the input layer configured as RGB+XYZ+D+F.

## 3.4 Recognition effect and analysis of multimodal feature fusion

The Fig 10 graphically depicts the recognition performance of the YOLOv5 model under the RGB+XYZ+D+F configuration, showing the recognition of apples under diverse light intensity conditions, including scenarios of occlusions, adhesions, and overlaps. The red rectangular boxes in the figure indicate the predicted locations of the fruits. Upon analysis of Fig 10, it is evident that the refined YOLOv5 model exhibits superior recognition and localization

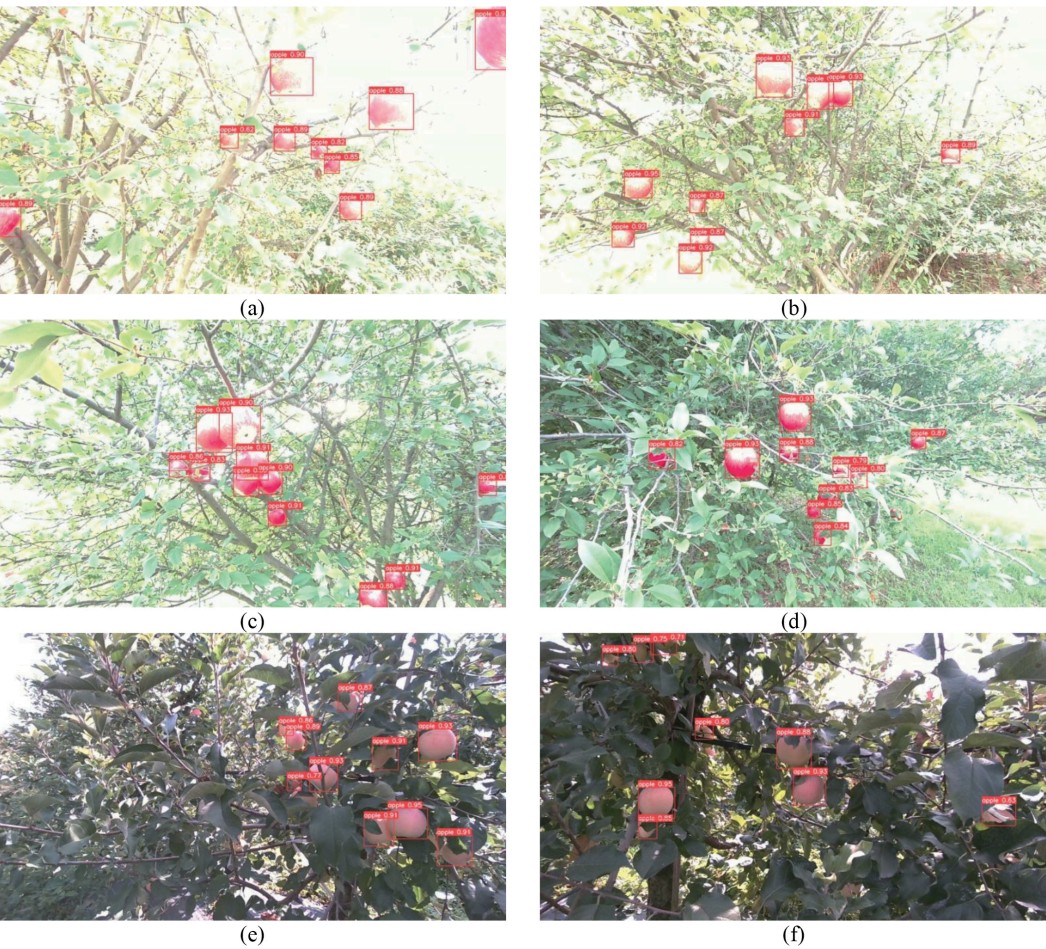

**Fig 10. Recognition performance of YOLOv5 under RGB+XYZ+D+F configuration.** (a) Apple prediction plot under lighting condition a, (b) Apple prediction plot under lighting condition b, (c) Apple prediction plot under lighting condition c, (d) Apple prediction plot under lighting condition d, (e) Apple prediction plot under lighting condition e, (f) Apple prediction plot under lighting condition f.

capabilities compared to all other models in the experiment. The model accurately detects and precisely locates the positions of the majority of visible fruits under diverse light intensity conditions, including scenarios with occlusions, adhesions, and overlaps.

## 4 Conclusion

This research introduces an innovative approach for the multimodal fusion of fruit features, significantly enhancing the accuracy of fruit detection without substantially modifying the underlying architecture of deep learning models. By integrating RGB images, key fruit features, depth information, and point cloud coordinates at the input layer, the model demonstrates superior performance in complex environments, as this combination leverages the complementary strengths of each modality: RGB and key features provide discriminative visual cues, while depth and point cloud data supply spatial-structural context to compensate for single-modal limitations. Experimental results indicate that the proposed method achieves a recognition accuracy of 95.8%, a recall rate of 96%, and an F1 score of 95.9% under complex lighting conditions with varying intensities. This research not only introduces a new technical approach for picking fruit but also serves as a valuable reference for the application of deep learning in multi-source data fusion. The primary contributions and conclusions of this research are summarized as follows:

1. **Innovative Multimodal Fusion Method:** Considering the constraints of traditional images in processing features within complex environments, this research integrates artificial features, such as color and edge information, to extract key fruit features for reducing information redundancy and interference. Furthermore, a new multimodal fusion method was proposed that combines key fruit features with depth information and point cloud coordinates. This approach enhances the model's robustness to varying lighting and background noise, strengthening its perceptive capabilities for fruit features.

2. **Improved YOLOv5 Model:** The input layer of the YOLOv5 model was optimized, introducing multi-channel input to enhance the network's recognition ability for fruit features.

3. **High-Accuracy Recognition Performance:** The model demonstrated superior performance in complex lighting conditions with varying intensities, achieving 95.8% accuracy, 96% recall, and an F1 score of 95.9% in apple recognition tasks. Specifically, it enhances the efficiency of robotic harvesting and lowers labor costs in agricultural applications.

4. **Future Research Directions:** Despite the notable achievements of this research, there is still room for further improvement and optimization. Future research can explore more efficient feature fusion strategies and how to apply these strategies to a broader range of agricultural automation scenarios.

In conclusion, the YOLOv5-based multimodal fusion approach presented in this study shows considerable potential in apple recognition. This research suggests that the proposed methodology could provide a robust foundation for enhancing precision agriculture and will pave the way for further exploration in the field of deep learning-based multi-source data fusion.

## Author contributions

**Conceptualization:** Xiaoyang Liu, Chongyang Hu.

**Data curation:** Chongyang Hu, Xupeng Huang, Chenxin Sun, Rongjin Zhu, Cheng Wang.

**Formal analysis:** Chongyang Hu, Rongjin Zhu.

**Funding acquisition:** Xiaoyang Liu, Chengzhi Ruan.

**Investigation:** Xiaoyang Liu, Chongyang Hu, Xupeng Huang, Chenxin Sun, Rongjin Zhu, Cheng Wang.

**Methodology:** Xiaoyang Liu, Chongyang Hu.

**Project administration:** Xupeng Huang, Yuxiang Zhang, Qian Shen, Hongbiao Zhou.

**Resources:** Xiaoyang Liu, Yuxiang Zhang, Qian Shen, Chengzhi Ruan.

**Software:** Chongyang Hu.

**Supervision:** Xiaoyang Liu, Yuxiang Zhang, Qian Shen, Hongbiao Zhou, Chengzhi Ruan.

**Validation:** Xiaoyang Liu, Chongyang Hu, Xupeng Huang, Chenxin Sun, Hongbiao Zhou.

**Visualization:** Chongyang Hu, Cheng Wang.

**Writing – original draft:** Xiaoyang Liu, Chongyang Hu.

**Writing – review & editing:** Xiaoyang Liu, Chongyang Hu.

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
