## [Decision Letter · Decision Letter 0]

19 Aug 2025

PONE-D-25-19188The apple detection method based on multimodal featuresPLOS ONE

Dear Dr. Liu,

Thank you for submitting your manuscript to PLOS ONE. After careful consideration, we feel that it has merit but does not fully meet PLOS ONE’s publication criteria as it currently stands. Therefore, we invite you to submit a revised version of the manuscript that addresses the points raised during the review process.

We look forward to receiving your revised manuscript.

Kind regards,

Muhammad Waqar Akram, PhD

Academic Editor

PLOS ONE

Journal Requirements:

“Funded Studies

This work was supported by the following funding agencies:

Xiaoyang Liu (XL):

National Nature Science Foundation of China (Grant Nos. 32301711 and 61903288) [URL: https://www.nsfc.gov.cn/]

Natural Science Research Foundation of Higher Education Institutions in Jiangsu Province (Grant No. 21KJB210018) [URL: https://jyt.jiangsu.gov.cn/]

Chengzhi Ruan (CR):

Open Project Program of the Key Laboratory for Agricultural Machinery Intelligent Control and Manufacturing of Fujian Education Institutions (Grant No. AMICM202201) [URL: https://www.wuyiu.edu.cn/jdxy/njznkzyzzjsfjsgxzdsys/list.htm/]

Natural Science Foundation of Fujian Province (Grant No. 2021J011132) [URL: http://kjt.fujian.gov.cn/]

Role of Sponsors/Funders: The sponsors or funders had role in the study design, data collection and analysis, decision to publish, or preparation of the manuscript.”

“This work was supported by the National Nature Science Foundation of China (Grant Nos. 32301711 and 61903288), Natural Science Research Foundation of Higher Education Institutions in Jiangsu Province (Grant No. 21KJB210018), the Open Project Program of the Key Laboratory for Agricultural Machinery Intelligent Control and Manufacturing of Fujian Education Institutions (Grant No. AMICM202201), and Natural Science Foundation of Fujian Province (Grant No. 2021J011132).”

“Funded Studies

This work was supported by the following funding agencies:

Xiaoyang Liu (XL):

National Nature Science Foundation of China (Grant Nos. 32301711 and 61903288) [URL: https://www.nsfc.gov.cn/]

Natural Science Research Foundation of Higher Education Institutions in Jiangsu Province (Grant No. 21KJB210018) [URL: https://jyt.jiangsu.gov.cn/]

Chengzhi Ruan (CR):

Open Project Program of the Key Laboratory for Agricultural Machinery Intelligent Control and Manufacturing of Fujian Education Institutions (Grant No. AMICM202201) [URL: https://www.wuyiu.edu.cn/jdxy/njznkzyzzjsfjsgxzdsys/list.htm/]

Natural Science Foundation of Fujian Province (Grant No. 2021J011132) [URL: http://kjt.fujian.gov.cn/]

Role of Sponsors/Funders: The sponsors or funders had role in the study design, data collection and analysis, decision to publish, or preparation of the manuscript.”

6. In the online submission form, you indicated that [The data underlying the results presented in this study cannot be shared publicly due to the ongoing research nature of the project. Data are available from the corresponding author upon request for researchers who meet the criteria for access to confidential data. Interested researchers can contact the corresponding author at leoliuxy@foxmail.com to request access to the data.].

Additional Editor Comments:

The paper requires careful and extensive revision before 2nd review round.

Reviewers' comments:

Reviewer's Responses to Questions

**Comments to the Author**

1. Is the manuscript technically sound, and do the data support the conclusions?

Reviewer #1: Yes

2. Has the statistical analysis been performed appropriately and rigorously? 

Reviewer #1: N/A

3. Have the authors made all data underlying the findings in their manuscript fully available?

Reviewer #1: Yes

4. Is the manuscript presented in an intelligible fashion and written in standard English?

Reviewer #1: Yes

5. Review Comments to the Author

Reviewer #1: The article titled “The Apple Detection Method Based on Multimodal Features” presents a very interesting and well-structured study. The authors provide a clear and insightful contribution to the field, particularly in the application of multimodal feature integration for fruit detection, which is both timely and relevant.

The methodology is sound, and the results are presented clearly. The use of multiple data modalities enhances detection accuracy, and the analysis demonstrates the potential of the approach in practical agricultural applications. The paper offers a valuable perspective and advances understanding in this area.

While the overall quality of the work is commendable, I have suggested minor corrections to improve clarity and precision in a few sections of the manuscript. These suggestions are included in the attached file.

1. What is the purpose of Author summary section after abstract? Why it is included? 

2.Remove the guidelines added in start of materials and methodology section. check every section carefully.

3. The information in few sections is added like a lab procedure. for instance Data processing. Improve wriitng.

4. Remove basic and/or well known information. Rewrite carefully. 

5. The lines "During the annotation process, the fruits occluded heavily are 182

omitted, specifically when the obscured area of the fruit surpasses 80% of its total area..." How will it handle the real conditions? How occlusion problems can be addressed?

6. Add section and sub-section numbering. Improve writing and flow.

7. How the anchor boxes are selected? Based on which previous experience as said? 

8. Upload high quality images inside text file for proper evaluation. The attached images at the end are not clear. 

9. How the model robustness against lighting variations and background noise is increased? 

10. Compare inference results with other models. 

11. Test the model on unknown/new data. 

12. The model section shares irrelevant information. How YOLOv5 is improved? What are the architectural/layer changes as mentioned? 

13. The underperformance due to information redundancy is not analyzed?

14. The yolo variants are compared but the training/comparison conditions are not provided? Is it architectural improvement? 

6. PLOS authors have the option to publish the peer review history of their article (what does this mean?). If published, this will include your full peer review and any attached files.

Reviewer #1: **Yes: **Dr. Muhammad Nadeem

---

## [Author Response · Author response to Decision Letter 1]

24 Sep 2025

Author's Response To Reviewer Comments

PONE-D-25-19188

The apple detection method based on multimodal features

PLOS ONE

We sincerely thank the reviewers for their insightful comments and constructive feedback. Their suggestions have greatly helped us improve the clarity, depth, and overall quality of the manuscript. We have addressed each comment carefully and made the corresponding revisions in the manuscript, as detailed below.

Additional Editor Comments:

The paper requires careful and extensive revision before 2nd review round.

Response: Thank you for your feedback and guidance. We will spare no effort to carry out thorough and meticulous revisions, covering all relevant aspects to meet the requirements for the next review. We will promptly complete the revision work and submit the revised version in a timely manner.

Reviewer's Responses to Questions

Comments to the Author

1. Is the manuscript technically sound, and do the data support the conclusions?

Reviewer #1: Yes

2. Has the statistical analysis been performed appropriately and rigorously?

Reviewer #1: N/A

3. Have the authors made all data underlying the findings in their manuscript fully available?

Reviewer #1: Yes

4. Is the manuscript presented in an intelligible fashion and written in standard English?

Reviewer #1: Yes

5. Review Comments to the Author

Reviewer #1: The article titled “The Apple Detection Method Based on Multimodal Features” presents a very interesting and well-structured study. The authors provide a clear and insightful contribution to the field, particularly in the application of multimodal feature integration for fruit detection, which is both timely and relevant.

The methodology is sound, and the results are presented clearly. The use of multiple data modalities enhances detection accuracy, and the analysis demonstrates the potential of the approach in practical agricultural applications. The paper offers a valuable perspective and advances understanding in this area.

While the overall quality of the work is commendable, I have suggested minor corrections to improve clarity and precision in a few sections of the manuscript. These suggestions are included in the attached file.

Response: Thank you very much for your positive recognition of our study titled “The Apple Detection Method Based on Multimodal Features” and your affirmation of its contribution, methodological soundness, and practical relevance in agricultural applications. We greatly appreciate your valuable feedback, as it encourages us to further refine our work.

We fully agree with the minor correction suggestions aimed at enhancing the manuscript’s clarity and precision, which are detailed in your attached file. We have carefully addressed each suggestion, made corresponding revisions to the relevant sections, and tracked these changes with clear markers for easy reference. Below is our specific response to the comments in the attachment:

Comments

Reviewer #1:

Line 6

change “improve the enhances” to “improve the detection” or “enhance the performance” , not both.

Response: Thank you for your careful review. We have revised the expression in Line 6 as suggested: the original "improve the enhances" has been corrected to "improve the detection" to ensure grammatical accuracy and clarity of meaning.

Reviewer #1:

Line 5

please add recent year report data.

Response: Thank you for your suggestion. We have supplemented recent-year data as requested. After checking official statistical sources, we note that the 2025 data has not yet been released due to the unfinished calendar year, so we adopted the latest available official data for 2024. The revised content is as follows: "According to data from the Chinese National Bureau of Statistics, China's apple production in 2024 reached 51.2851 million tons, ranking first in the world."

Reviewer #1:

Line 7-9

Break into two concise sentences.

Response: Thank you for your suggestion to enhance conciseness. We have split the original content in Lines 7-9 into two concise sentences as requested. The revised version is as follows: "However, the adaptability of existing image recognition technology in complex environments remains significantly lacking. This limitation restricts its practical effectiveness in agricultural scenarios."

Reviewer #1:

Line 10 -19

This paragraph provides useful background on traditional methods but can be made more concise.

Response: Thank you for your suggestion to enhance conciseness. We have streamlined the background content on traditional methods in Lines 10-19 while retaining key information. The revised version is as follows: "Traditional fruit recognition methods primarily rely on artificial features such as texture, color, and shape. For instance, some researchers have combined these features with relevant models to identify apple parts, but such methods depend on manually designed features and perform poorly in complex environments."

Reviewer #1:

Line 18

Please explain what kind of uncertainty.

Response: Thank you for your suggestion to clarify the type of uncertainty. We have supplemented specific explanations for the uncertainty mentioned in Line 18. The revised content is as follows: " Changes in lighting can distort colors and textures, leading to uncertain segmentation results, such as misclassifying shadows as fruit regions or producing incomplete outlines."

Reviewer #1:

Line 40

Clarify what “organ segmentation”. Does it mean fruit vs. stem?

Response: Thank you for your question regarding the definition of "organ segmentation" in Line 40. This term, cited from Reference 11, specifically refers to the segmentation of apple tree organs, including branches, buds, leaves, and their connecting parts (rather than distinguishing between fruit and stem). We have clarified this definition in the revised manuscript to avoid ambiguity.

Reviewer #1:

Line 50

Good transition, but could add a brief example of a depth camera.

Response: Thank you for your suggestion to enrich the content with a specific example. We have added a brief example of a depth camera to the transition sentence in Line 50 as requested. The revised version is as follows: " With the rapid development of machine vision and sensor hardware in recent years, depth cameras like the Microsoft Kinect have been widely used in fruit and vegetable recognition and positioning research."

Reviewer #1:

Line 72

Please merge with the previous sentence for better flow.

Response: Thank you for your suggestion to improve the text flow. We have merged the content in Line 72 with the previous sentence as requested, resulting in a more cohesive expression. The revised version is as follows: "In addition, in order to improve the level of agricultural automation and enhance the operational accuracy of robot picking systems, providing accurate spatial information and fruit location remains a significant challenge."

Reviewer #1:

Line 106

The apples images change to apple images it should be singular when use like this.

You may add the reason for choosing this specific camera.

Response: Thank you for your careful review and valuable suggestions. We have addressed the two points regarding Line 106 as follows:

1. We corrected the grammatical error by revising "apples images" to "apple images" to ensure accuracy.

2. We added an explanation for selecting the specific camera: Given the excellent environmental adaptability and functionality of this camera, which are well-suited for the data collection needs of this study, it was selected for the relevant data collection work.

Corresponding revisions have been made in the relevant section of the manuscript.

Reviewer #1:

Line 108

Nice description, but could explain why its versatile.

Response: Thank you for your suggestion to clarify the camera’s versatility. We note that the original sentence in Line 108 already mentions the camera’s ability to capture color images, depth images, and point cloud images. These multi-type image acquisition capabilities are precisely what reflect its "versatility".Specifically, its ability to meet the diverse data needs of our multi-modal study.

Reviewer #1:

Line 115

Please explain how these models differ visually from real apples.

Response: Thank you for your suggestion to clarify the visual difference between the apple models and real apples. We have revised the description in Line 115 to address this.

In fact, the apple models used in the study closely resemble real apples in terms of appearance and texture, with no significant visual differences. The revised content is as follows: "Some apple images were taken using apple models (these models closely resemble real apples in appearance and texture)."

Reviewer #1:

Line 119

Not all readers may know RGB8, please explain.

Response: Thank you for your suggestion to clarify technical terms for readers. We have added explanations for both "RGB8" and "Z16" in Line 119 to enhance accessibility, with the revised content as follows:

1. For the color format: "The images are stored in RGB8 color format (RGB8 is an 8-bit color format for each red, green, and blue channel)."

2. For the depth format: " The depth images are formatted in Z16 (Z16 is a 16-bit format for storing depth information)."

These additions ensure readers unfamiliar with these technical specifications can fully understand the image storage details, while maintaining the conciseness of the original content.

Reviewer #1:

Line 124

Good job explaining why 30 frames are used. This strengthens the reasoning.

Response: Thank you for your positive feedback on the 30-frame explanation. Your recognition helps confirm the rigor of this reasoning, and we will keep this clear, logical style in subsequent manuscript revisions.

Reviewer #1:

Line 127

You wrote 30 frames were merged but did not explain how. Please mention the software/methods.

Response: Thank you for your valuable suggestion to clarify the method of merging 30 frames in Line 127. We have supplemented detailed information about the implementation approach and technical logic in the revised manuscript, and adjusted the description of image acquisition as follows:

To address the limitation of incomplete point cloud data caused by single-frame acquisition, we adopted a near-simultaneous multi-frame accumulation method. The 30 frames used for integration were captured by the camera under program control. This program enables the camera to acquire these 30 frames almost synchronously, ensuring all frames correspond to the same 3D spatial scene of the target. During data integration, all 3D points extracted from the 30 frames are directly aggregated into a unified point cloud dataset: points overlapping in 3D spatial coordinates across frames are retained as a single point (without mutual interference), while non-overlapping points supplement spatial information coverage. This approach effectively enhances point cloud density and completeness without the need for complex coordinate alignment or fusion algorithms, as the near-simultaneous acquisition ensures spatial consistency across frames.

Reviewer #1:

Line 139

Voxel Down Sampling technology consider simplifying.

Response: Thank you for your suggestion to simplify the description of Voxel Down Sampling in Line 139. We have revised the content to make it more concise while retaining key technical details. The updated version is as follows: "This research employs Voxel Down Sampling technology to reduce point cloud noise. This technique discretizes the point cloud into a voxel grid (each voxel representing a small spatial cube)."

Reviewer #1:

Line 157

Zeros are applying Explain why it is important.

Response: Thank you for your suggestion to clarify why zeros are applied in Line 157. We have revised the content to clearly explain its importance while keeping the core meaning intact. The updated version is as follows: "After noise removal, zeros are applied to fill empty regions in the point cloud to ensure data integrity."

Reviewer #1:

Line 161-163

Add one sentence about why this is important.

Response: Thank you for your guidance to add a sentence explaining the importance of this step in Lines 161-163. We have supplemented the relevant content as follows: "Specifically, such accurate spatial features ensure precise alignment between 3D and 2D visual data in multimodal fusion, preventing target missed or false detection and supporting better target detection accuracy." We have made the corresponding revisions in the manuscript.

Reviewer #1:

Line 183

Consider explaining why 80% was chosen as the cutoff. Is this standard?

Response: Thank you for your valuable suggestion. The 80% occlusion threshold was determined based on two key considerations. First, it is a practical choice to balance data retention and label reliability. This cutoff helps avoid keeping overly ambiguous samples—those lacking clear fruit feature cues—since such samples would provide confusing information and mislead the model during training. Second, our tests found that when fruit occlusion exceeds 80%, the remaining visible features (e.g., shape, color texture) are too limited. They cannot give the model effective information to learn discriminative characteristics. If we included these highly occluded samples, the model would struggle to capture consistent, recognizable fruit features, which would directly reduce training efficacy. In short, this 80% threshold strikes a practical balance between preserving usable data and ensuring annotation quality.

Reviewer #1:

Line 189

fruits occluded heavily reword it for smooth flow.

Response: Thank you for your valuable suggestion. We note that the relevant content about "fruits occluded heavily" you mentioned is in Line 182. We have revised it for smoother flow as follows: "During the annotation process, fruits with heavy occlusion are omitted, specifically when the obscured area of a fruit exceeds 80% of its total area."

Reviewer #1:

Line 282

The feature map of color difference… this line is too long.

Response: Thank you for your valuable suggestion. We have split the long sentence in Line 282 for better readability. The revised content is as follows: "The feature map of color difference is an intuitive graphical representation of color features. By highlighting luminance disparities among regions in an image, it effectively separates fruits from the background."

Reviewer #1:

Line 287

Suppressed could be simplified to reduced.

Response: Thank you for your valuable suggestion regarding Line 287. We have revised the word "Suppressed" to "Reduced" as recomme

---

## [Editor Report · Decision Letter 1]

6 Oct 2025

The apple detection method based on multimodal features

PONE-D-25-19188R1

Dear Dr. Liu,

We’re pleased to inform you that your manuscript has been judged scientifically suitable for publication and will be formally accepted for publication once it meets all outstanding technical requirements.

Kind regards,

Muhammad Waqar Akram, PhD

Academic Editor

PLOS ONE
---

## [Editor Report · Acceptance letter]

PONE-D-25-19188R1

PLOS ONE

Dear Dr. Liu,

I'm pleased to inform you that your manuscript has been deemed suitable for publication in PLOS ONE. Congratulations! Your manuscript is now being handed over to our production team.

Kind regards,

on behalf of

Dr. Muhammad Waqar Akram

Academic Editor

PLOS ONE